# Assessment of the Rheological and Mechanical Properties of Geopolymer Concrete Comprising Fly Ash and Fluid Catalytic Cracking Residue as Aluminosilicate Precursor

**Tuan Anh Le [1,2], Sinh Hoang Le [3,4], Thuy Ninh Nguyen [1,2] and Khoa Tan Nguyen [3,5,*]**

1   Faculty of Civil Engineering, Ho Chi Minh City University of Technology, 268 Ly Thuong Kiet Street, District 10, Ho Chi Minh City 700000, Vietnam; latuan@hcmut.edu.vn (T.A.L.); ninhthuy@vnuhcm.edu.vn (T.N.N.)
2   Vietnam National University Ho Chi Minh City, Linh Trung Ward, Thu Duc District, Ho Chi Minh City 700000, Vietnam
3   Institute of Research and Development, Duy Tan University, Da Nang 550000, Vietnam; lehoangsinh@duytan.edu.vn
4   Faculty of Natural Science, Duy Tan University, Da Nang 550000, Vietnam
5   Faculty of Civil Engineering, Duy Tan University, Da Nang 550000, Vietnam
*   Correspondence: nguyentankhoa@duytan.edu.vn; Tel.: +84-82-927-0589

**Featured Application: The geopolymer concrete comprising fly ash and fluid catalytic cracking residue could be applied for producing eco-friendly precast concrete products in Vietnam, which are precast wall, panel, or brick.**

**Abstract:** The use of fluid catalytic cracking (FCC) by-products as aluminosilicate precursors in geopolymer binders has attracted significant interest from researchers in recent years owing to their high alumina and silica contents. Introduced in this study is the use of geopolymer concrete comprising FCC residue combined with fly ash as the requisite source of aluminosilicate. Fly ash was replaced with various FCC residue contents ranging from 0–100% by mass of binder. Results from standard testing methods showed that geopolymer concrete rheological properties such as yield stress and plastic viscosity as well as mechanical properties including compressive strength, flexural strength, and elastic modulus were affected significantly by the FCC residue content. With alkali liquid to geopolymer solid ratios (AL:GS) of 0.4 and 0.5, a reduction in compressive and flexural strength was observed in the case of geopolymer concrete with increasing FCC residue content. On the contrary, geopolymer concrete with increasing FCC residue content exhibited improved strength with an AL:GS ratio of 0.65. Relationships enabling estimation of geopolymer elastic modulus based on compressive strength were investigated. Scanning electron microscope (SEM) images and X-ray diffraction (XRD) patterns revealed that the final product from the geopolymerization process consisting of FCC residue was similar to fly ash-based geopolymer concrete. These observations highlight the potential of FCC residue as an aluminosilicate source for geopolymer products.

**Keywords:** geopolymer concrete; fly ash; FCC residue; rheology; mechanical properties; microstructure; SEM; XRD

## 1. Introduction

Geopolymers are inorganic polymers produced by combinations of aluminosilicate precursors and alkali activators at ambient or elevated temperatures. The term geopolymer, introduced for the first time by Davidovits in 1979 [1], typically implies binders with zero or low levels of cement that negate environmental issues associated with conventional Portland cement production, such as high embodied carbon. The carbon footprint of geopolymer material manufacture has been reported to be 43% of that for conventional

cement [2]. In addition, geopolymer materials are known to exhibit outstanding mechanical [3–5], durability [6,7], and shrinkage properties [8], as well as chemical erosion resistance [9].

The main constituent materials of geopolymer materials include aluminosilicate precursors sourced from industrial by-products and alkali solution comprising alkali hydroxide (potassium or sodium hydroxide) and alkali silicate (potassium or sodium silicate). The availability of aluminosilicate by-products such as fly ash, ground-granulated blast-furnace slag (GGBS), red mud, and metakaolin is dependent on local resources [10]. Recently, the supply of fly ash (one of the most commonly used and cost-effective binder materials for geopolymers) has decreased due to a reduction in coal-fired power plants related to increasing uses of renewable energy sources such as wind and solar [11]. Regional shortages have been reported in North America and European countries [12]. For instance, the fly ash shortage in the U.K. is anticipated to be approximately 2 million tons in 2030 due to the planned closure of all coal-fired power plants by 2025 [12,13].

Against this background, by-products from fluid catalytic cracking (FCC) processes within the oil refining industry are proven to offer a potential alternative to conventional aluminosilicate fly ash [14–17]. The silica and alumina content of FCC by-products is typically greater than 90% by mass [14], a feature that has been successfully exploited in their use as supplementary cementitious materials in conventional Portland cement concrete binders [17]. Contrary to the downward trend of coal-fired power plants described above, production rates from the petroleum industry have remained steady in oil-producing countries such as Vietnam, the United Kingdom, and the United States [18–20]. For example, production of crude oil in the U.K. increased from 12 million tons in 2017 to over 13 million tons in the second quarter of 2020 [19]. The corresponding amount of FCC by-product sent to landfills each year was reported to be approximately 840 thousand tons [21].

According to Rodríguez et al. [15], the spent FCC catalyst showed great practical potential as an aluminosilicate precursor for geopolymer production. In this work, its high reactivity was established from observations from X-ray diffraction, Fourier transform infrared spectroscopy, solid-state $^{29}$Si and $^{27}$Al magic angle spinning nuclear magnetic resonance, and scanning electron microscopy (SEM). Ruiz et al. [16] synthesized geopolymer binders using spent an FCC catalyst and alkali activator comprising sodium silicate and sodium hydroxide (NaOH) with various levels of NaOH concentration and silica modulus. The resulting geopolymer binders achieved a compressive strength of 25 MPa after 7 days. In a study by Tashima et al. [17], FCC residue-based geopolymer mortar specimens with ratios of $SiO_2/Na_2O$ ranging from 0–1.46 were cured under high-humidity conditions (relative humidity of 100%) at 65 °C for three days. The resulting compressive and flexural strengths varied from 8.5–68 MPa and 2.5–11.5 MPa, respectively. Microstructurally, the FCC geopolymer mortar specimens were observed to be in a dense-compact amorphous state via SEM analysis. In addition to geopolymer precursor applications, FCC residue and slurry oil have been used as the main constituent materials in conventional concrete [22,23], soil stabilization [24], ceramic [25], and paving asphalt [26]. Thorough research of relevant literature yielded that minimal comprehensive research has been undertaken to investigate the effects of fly ash-FCC combinations on geopolymerization processes and the mechanical properties of geopolymer concrete.

As such, this study aims to assess the rheological and hardened properties of geopolymer concrete containing FCC residue as a full or partial replacement of fly ash. Effects of FCC residue content (0, 20, 40, 60, 80, and 100% by mass) on yield stress, plastic viscosity, compressive strength, elastic modulus, and flexural strength of geopolymer concrete were investigated using standard mechanical testing. Additionally, SEM and X-ray diffraction (XRD) techniques were employed to evaluate the roles of FCC residue on microstructural properties of geopolymer concrete. Image processing using ImageJ software was used to determine void volume percentage in binary SEM photos. This method has proven its ability to detect microstructures of concrete and rocks in several studies [27–29]. The novelty of current research is the development of FCC residue-based geopolymer concrete

that can be partially or fully replaced by fly ash-based geopolymer concrete for producing precast geopolymer concrete.

## 2. Materials and Experimental Methods

### 2.1. Materials

The main constituent materials investigated for geopolymer concrete manufacture in this study included low-calcium fly ash (class F), FCC residue, alkali liquid, and fine and coarse aggregates. Low-calcium fly ash was provided by Formosa Dong Nai Thermal Power Plant (Dong Nai, Vietnam). Coarse FCC was obtained from Petro Vietnam-Dung Quat (Vietnam). The densities and particle size ranges for the low-calcium fly ash and FCC residue were 2.5 and 0.88 g/cm$^3$, and 5–10 and 18–100 μm, respectively. Coarse FCC residue was initially milled and dried at 100 °C for 24 h to obtain finer powder fractions (18–100 μm) and promote higher levels of reactivity [16]. Representative microscopic images of the fly ash and FCC residue particles are provided in Figure 1, which clearly highlights the disparity in particle sizes. Apparent from this figure is particle shape and texture, which for the FCC residue is marginally more angular and rougher relative to the spherical, smooth fly ash particles. In contrast to these physical differences, both materials had comparable SiO$_2$ and Al$_2$O$_3$ contents, as presented in Table 1.

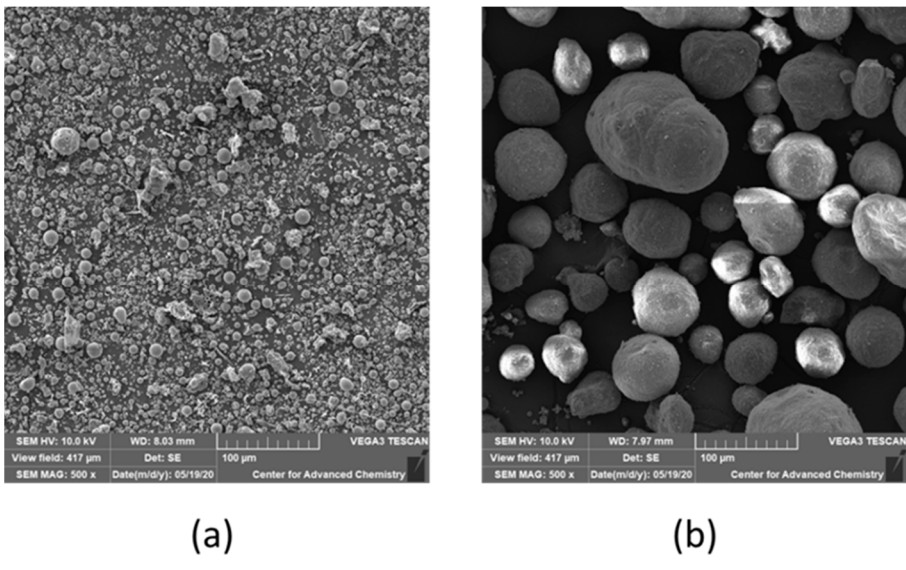

(a)                                              (b)

**Figure 1.** SEM images of (**a**) fly ash and (**b**) FCC residue particles.

**Table 1.** Chemical compositions (% by mass) of fly ash and fluid catalytic cracking (FCC) residue.

| Materials | SiO$_2$ | Al$_2$O$_3$ | Fe$_2$O$_3$ | CaO | K$_2$O & Na$_2$O | MgO | SO$_3$ | LOI * |
|---|---|---|---|---|---|---|---|---|
| Fly ash | 51.7 | 31.9 | 3.48 | 1.21 | 1.02 | 0.81 | 0.25 | 9.63 |
| FCC residue | 55 | 39 | 0.38 | 0.5 | 0.21 | - | <1 | - |

\* LOI: Loss on ignition.

The alkali liquid (AL) used in this study consisted of sodium silicates (8.37% Na$_2$O, 27.63% SiO$_2$, and 64% H$_2$O by mass) and sodium hydroxide (NaOH, concentration of 10 M) at a ratio of 1:1 by mass, which was purchased from VMC Group (Vietnam). Local sand with a density of 2.65 g/cm$^3$ and crushed rock with a maximum size of 20 mm and density of 2.70 g/cm$^3$ were used as fine and coarse aggregates, respectively.

### 2.2. Mixture Proportioning

Mixture proportion details of the geopolymer concrete investigated in this study are shown in Table 2. Mix design was based on three broad groups of six mixes (G1–6, G7–12,

and G13–18) with increasing binder contents of 300, 400, and 500 kg/m$^3$, respectively. For each group, the alkali content was held constant at approximately 100 kg/m$^3$. As such, corresponding ratios of alkali liquid (AL) to geopolymer source (GS) (referred to as AL:GS) for each group were 0.65, 0.50, and 0.40, respectively. Within each mix group, six binder combinations were considered based on the following FCC residue replacement levels for fly ash: 0, 20, 40, 60, 80, and 100% by mass.

**Table 2.** Mix design of geopolymer concrete.

| Mix ID | Mix Design Parameters | | Mass of Constituent Materials (kg) | | | | | |
|---|---|---|---|---|---|---|---|---|
| | AL:GS | FCC Residue (%) | CA | FA | Fly Ash | FCC Residue | Sodium Silicate | Sodium Hydroxide |
| G1 | 0.65 | 0 | 1098 | 810 | 300 | 0 | 97.5 | 97.5 |
| G2 | 0.65 | 20 | 1098 | 810 | 240 | 60 | 97.5 | 97.5 |
| G3 | 0.65 | 40 | 1098 | 810 | 180 | 120 | 97.5 | 97.5 |
| G4 | 0.65 | 60 | 1098 | 810 | 120 | 180 | 97.5 | 97.5 |
| G5 | 0.65 | 80 | 1098 | 810 | 60 | 240 | 97.5 | 97.5 |
| G6 | 0.65 | 100 | 1098 | 810 | 0 | 300 | 97.5 | 97.5 |
| G7 | 0.5 | 0 | 1063 | 770 | 400 | 0 | 100 | 100 |
| G8 | 0.5 | 20 | 1063 | 770 | 320 | 80 | 100 | 100 |
| G9 | 0.5 | 40 | 1063 | 770 | 240 | 160 | 100 | 100 |
| G10 | 0.5 | 60 | 1063 | 770 | 160 | 240 | 100 | 100 |
| G11 | 0.5 | 80 | 1063 | 770 | 80 | 320 | 100 | 100 |
| G12 | 0.5 | 100 | 1063 | 770 | 0 | 400 | 100 | 100 |
| G13 | 0.4 | 0 | 950 | 760 | 500 | 0 | 100 | 100 |
| G14 | 0.4 | 20 | 950 | 760 | 400 | 100 | 100 | 100 |
| G15 | 0.4 | 40 | 950 | 760 | 300 | 200 | 100 | 100 |
| G16 | 0.4 | 60 | 950 | 760 | 200 | 300 | 100 | 100 |
| G17 | 0.4 | 80 | 950 | 760 | 100 | 400 | 100 | 100 |
| G18 | 0.4 | 100 | 950 | 760 | 0 | 500 | 100 | 100 |

*2.3. Specimen Preparation*

Sodium hydroxide in solid form was initially mixed with water, and the solution was then mixed with sodium silicate to prepare the required alkali liquid. Following suggestions from a study by Davidovits [30], alkali liquid was prepared one day before mixing with other geopolymer constituents to promote better polymerization. Geopolymer concrete specimens were produced by mixing fly ash, FCC residue, alkali liquid, and fine and coarse aggregates in the proportions provided in Table 2. Firstly, fly ash and FCC residue were mechanically blended for approximately three minutes. Alkali liquid was then added and mixed for four minutes. Finally, fine and coarse aggregates were added to the slurry and mixed for ten minutes. Fresh geopolymer concrete was cast in steel molds and cured in an oven at 60 °C for 4 h.

*2.4. Test Methods*

To assess the effects of FCC residue contents on rheological and hardened properties of geopolymer concrete, such as yield stress, plastic viscosity, elastic modulus, and compressive and flexural strength, testing methods in compliance with ASTM standards were carried out. Rheological properties were calculated based on the slump value obtained from the modified slump cone test [31], which is illustrated in Figures 2 and 3.

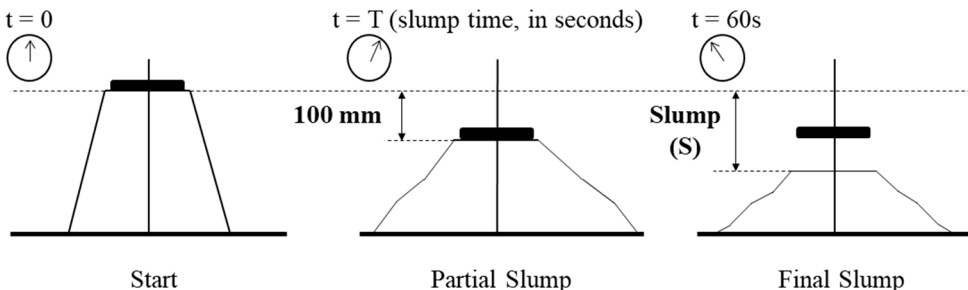

**Figure 2.** Schematics of the modified slump cone test [31].

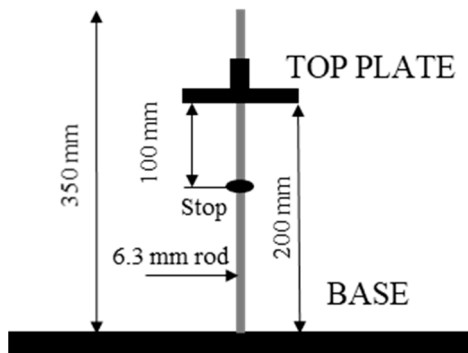

**Figure 3.** Device used for modified slump cone test [31].

Yield stress could be identified from the final slump, using the following equations [32]:

$$\tau_0 = \frac{\rho}{347}(300 - S) + 212 \tag{1}$$

where $\tau_0$ is yield stress (Pa); $\rho$ is geopolymer concrete density (kg/m$^3$); and $S$ is geopolymer concrete slump (mm). The viscosity was determined from the 100 mm slump time using an empirical formulation [31], which is:

$$\mu = \rho T \times 1.08 \times 10^{-3}(S - 175) \text{ for } 200 \text{ mm } < S < 260 \text{ mm} \tag{2}$$

$$\mu = 0.0025\rho T \text{ for } S < 200 \text{ mm} \tag{3}$$

where $\mu$ is plastic viscosity for geopolymer concrete (Pa·s) and $T$ is partial slump time (s). In this study, Equation (3) was used to calculate the plastic viscosity of geopolymer concrete because all values of slump are lower than 200 mm.

Elastic modulus was assessed after 7 days in accordance with ASTM C469 [33] by measuring stress and longitudinal strain of cylindrical geopolymer specimens (150 mm diameter × 300 mm height). Specimens were subjected to compressive load at a constant rate of 0.24 ± 0.03 MPa/s and secant modulus values recorded at a stress level of 40% of the corresponding average compressive cylinder strength. Compressive strength testing was performed on cylindrical specimens (100 mm diameter × 200 mm height) with loading rates ranging from 0.15 to 0.35 MPa/s in accordance with ASTM C39 [34]. Flexural strength testing was carried out on simply supported 100 × 100 × 400 mm beams with a span length of 300 mm in accordance with ASTM C78 [35]. Loading rates ranged from 0.86 to 1.21 MPa/min to the point of failure. For all strength tests, a group of four specimens was fabricated for one mix proportion. In total, 216 specimens were used for strength tests in this research.

In addition to physical and mechanical testing, scanning electron microscopy (SEM) VEGA 3 SBH (TESCAN) was used for morphological study. The samples used for the SEM test were picked up from broken pieces of specimens after testing. The X-ray diffractometer DE/D8 Advance, Bruker mark using CuKα radiation was used to perform XRD analyses.

The samples were examined in the range of 2θ, ranging from 10° to 60°. The XRD patterns were used to identify crystalline components in the fly ash/FCC residue geopolymer concrete samples.

## 3. Results and Discussion

### 3.1. Rheological Properties of Geopolymer Concrete

Figure 4 presents the rheological properties such as yield stress and plastic viscosity of fresh geopolymer concrete with FCC residue content ranging from 0–100%. In general, increasing content of FCC residue as replacement of fly ash led to increasing yield stress (1597–1948 Pa) and plastic viscosity (210–476 Pa·s). Fly ash-based geopolymer specimens (G1, G7, and G13) comprising only fly ash as source material were highly workable concrete with the lowest yield stress and plastic viscosity ranges of 1597–1671 (Pa) and 210–380 (Pa·s), respectively.

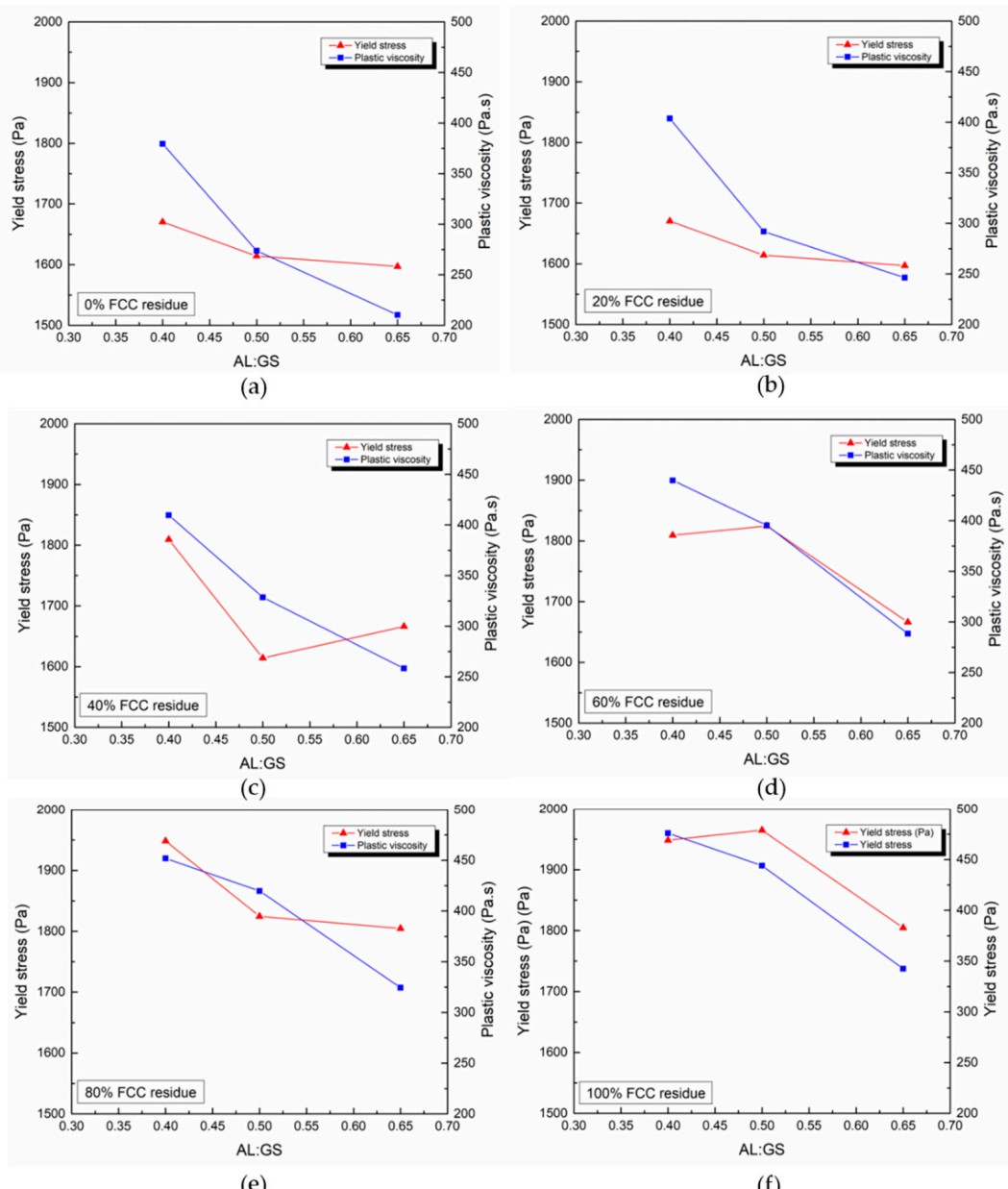

**Figure 4.** Relationship between yield stress and plastic viscosity of fresh geopolymer concrete and FCC residue content (% by binder mass). (**a**) 0% FCC residue. (**b**) 20% FCC residue. (**c**) 40% FCC residue. (**d**) 60% FCC residue. (**e**) 80% FCC residue. (**f**) 100% FCC residue.

For instance, in the case of AL:GS = 0.65 mixtures, yield stress and plastic viscosity values of G4–6 with high FCC residue content (≥60%) were 4–12% and 27–39% higher than G1 specimen without FCC residue, respectively. This can possibly be explained by the fact that fresh geopolymer concrete with both spherical-shaped fly ash (as shown in Figure 1a) and angular-shaped FCC residue (as shown in Figure 1b) possessed higher frictional resistance compared to geopolymer with only fly ash. A combination of particles with spherical and angular shapes made the mixture less workable than that with only spherical fly ash particles. In addition, the large particle size of FCC residue was one of the main causes of reduction in fresh geopolymer workability. This was in agreement with previous findings reported by Ekwulo and Eme [36].

### 3.2. Mechanical Properties of Hardened Geopolymer Concrete

The relationship between FCC residue content (% by binder mass) geopolymer concrete mechanical properties such as compressive strength and flexural strength after 7 curing days is shown in Figure 5. As shown in Figure 5a, geopolymer concrete containing fly ash and FCC residue possessed a wide range of mechanical properties, with compressive strength of 10–26 MPa and flexural strength of 0.9–2.8 MPa.

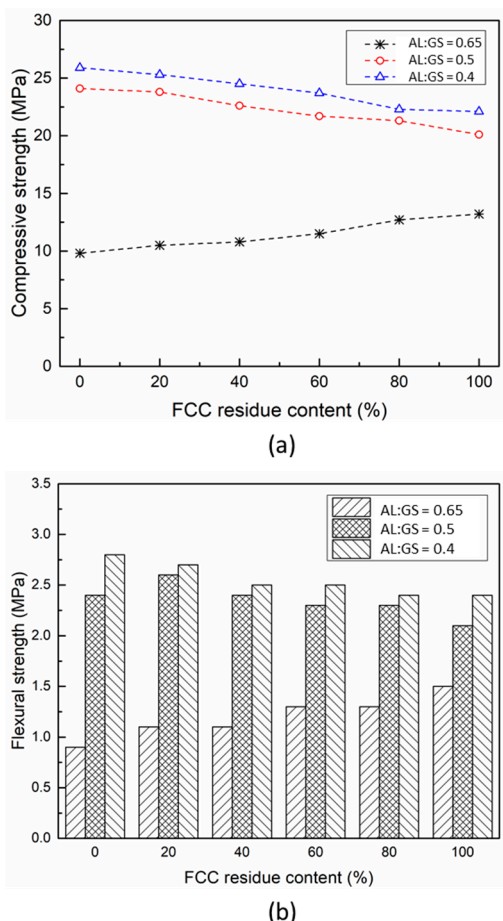

**Figure 5.** Relationship between FCC residue contents (% by binder mass) and mechanical properties of geopolymer concrete. (**a**) Compressive strength. (**b**) Flexural strength.

In the case of fly ash-based geopolymer concrete without FCC residue, geopolymer strength increased with a decrease in AL:GS ratio. There were two different trends presented in Figure 5, with the increased mechanical performance trend associated with increasing FCC residue content in geopolymer mixtures AL:GS = 0.65 and decreasing compressive strength associated with increasing FCC residue content in geopolymer mixtures AL:GS = 0.5 and AL:GS = 0.4. The positive performance of geopolymer concrete mixes

AL:GS = 0.65 comprising FCC residue was highlighted, with 7–34% higher compressive strength than fly ash-based geopolymer concrete. In contrast to mixes AL:GS = 0.65, there is a negative impact of FCC residue content on geopolymer mixes AL:GS = 0.5; and AL:GS = 0.4, with maximum 17% strength reduction associated with geopolymer concrete with 100% FCC residue. Additionally, when FCC residue content increased from 0 to 100%, the flexural strength of AL:GS = 0.65 mixtures increased to 67%, while up to 14% of the flexural reduction was observed in the cases of AL:GS = 0.65 mixtures.

Elastic modulus is a significant parameter of concrete for structural design, which is calculated as the secant modulus measured at the stress value equal to 40% of the average compressive strength of concrete cylinders, according to ASTM C469 [33]. Figure 6a presents the relationship between FCC residue contents (% by binder mass) and elastic modulus of geopolymer concrete. As shown in this figure, geopolymer concrete containing FCC residue possessed a modulus of elasticity ranging within 12.3–21.3 GPa. A similar trend with concrete strength obtained in Figure 5 was found where decreasing elastic modulus corresponded to increasing FCC residue content in geopolymer mixture AL:GS = 0.5 and 0.4, while increasing FCC residue content in geopolymer mixture AL:GS = 0.65 led to an improvement in elastic modulus. A comparison between elastic modulus obtained from experiments and estimated by the following equations [37–41] is shown in Figure 6b.

$$E_c = 3320\sqrt{f_c} + 6900 \tag{4}$$

$$E_c = 0.85 \times 2.15 \times 10^4 \times \sqrt[3]{0.1f_c} \tag{5}$$

$$E_c = 2707\sqrt{f_c} + 5300 \tag{6}$$

$$E_c = 0.037\rho^{1.5}\sqrt{f_c} \tag{7}$$

$$E_c = 5300\sqrt[3]{f_c} \tag{8}$$

where $f_c$ is the mean compressive strength (MPa) and $\rho$ is the unit-weight of geopolymer concrete (kg/m$^3$).

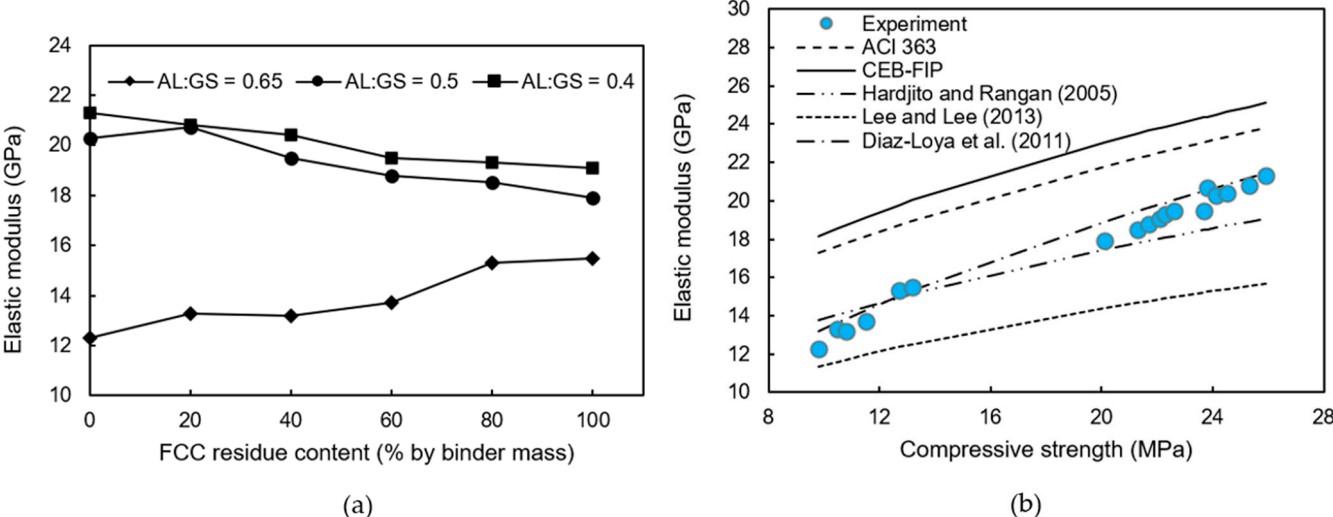

(a)          (b)

**Figure 6.** Relationship between: (**a**) FCC residue contents (% by binder mass) and elastic modulus; (**b**) elastic modulus and corresponding compressive strength.

The average unit-weight of geopolymer concrete was 2415 kg/m$^3$. It is noted that Equations (3) and (4) were used to calculate the value of elastic modulus of normal concrete, while Equations (5)–(7) are designed for the determination of elastic modulus of geopolymer concrete.

According to Figure 4, the value of elastic modulus estimated using equations for geopolymer concrete was lower than that of conventional concrete. The most significant

differences between experimental data and predicted data using Equations (3) and (4) were 32% to 40%, respectively. Therefore, those equations were not recommended to be used to predict the value of elastic modulus with given compressive strength. Equation (6) obtained from the study by Diaz-Loya et al. (2011) [40] showed the highest possibility to predict the modulus of elasticity with given compressive strength.

### 3.3. Microstructures of Geopolymer Concrete

XRD analysis results of geopolymer concrete specimens are shown in Figure 7. Most of the crystalline phases of quartz were detected as sharp peaks in all mixtures. Similar XRD patterns were observed for all geopolymer concrete mixes, indicating FCC residue can be used as a replacement of fly ash precursor in geopolymer concrete.

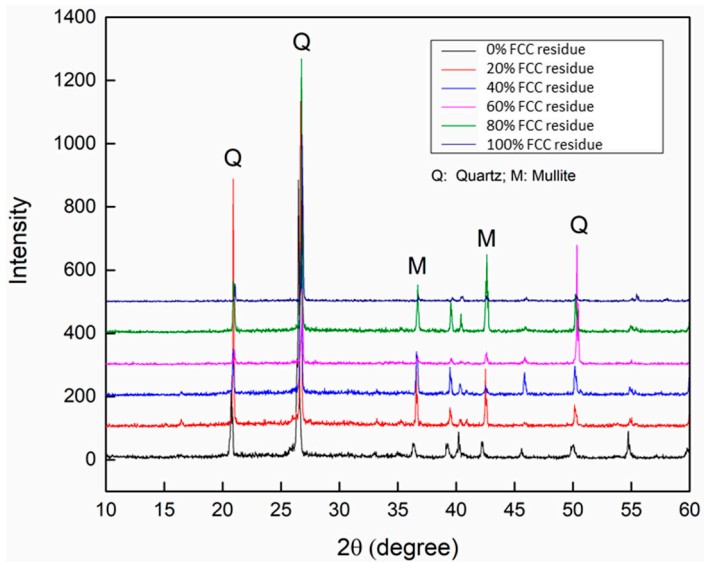

**Figure 7.** X-ray diffraction (XRD) patterns of geopolymer concrete with various FCC contents (% by binder mass).

To determine the differences between specimens consisting of fly ash and FCC residue and explain the results from previous sections, four geopolymer mixtures AL:GS = 0.65 (G1,5) and AL:GS = 0.5 (G7,11) were selected for SEM observations. The reason for choosing the aforementioned mixtures instead of AL:GS = 0.4 mixtures was that the 7-day compressive strength of AL:GS = 0.5 and AL:GS = 0.4 was similar. Results of microstructural observation of AL:GS = 0.65 (G1,5) and AL:GS = 0.5 (G7,11) specimens are illustrated in Figure 8, whereas Figure 9 shows void volume percentages of those specimens and their binary images using ImageJ at a threshold value at 76. Although image processing by ImageJ software is probably affected by the vagueness of the objects, such as aggregate particles and cementitious matrix, this technique provides an easy method to determine void percentage in geopolymer microstructures. These studies [42,43] in which fuzzy divergence was used in image detection are recommended for improvement of quality and quantity of image processing in future research. For specimen G1 (0% FCC residue), it was apparent that its microstructure with 29.7% void (by volume) was not as dense as specimen G5 (80% FCC residue) with 24.8% void by volume. However, for the case of AL:GS = 0.5 mixtures, the microstructure of the G7 (15.8%) mixture with only fly ash was denser than the G11 (22.7%) mixture with FCC residue. This might be possibly explained by the fact that for the mixtures with a low ratio of AL:GS (0.5), with the presence of FCC residue particles, the amount of alkali liquid was not enough for both fly ash and FCC residue particles to transport and arrange for producing geopolymer products. As a result, the compressive strength of hardened geopolymer concrete consisting of FCC residue (G11) was reduced due to the presence of a great number of large voids compared to fly ash-based geopolymer concrete (G7), as shown in Figure 8c,d. There were numerous unreacted FCC

residue particles and voids inside the concrete structure in Figure 8d. On the contrary, AL:GS = 0.65 mixtures with a high ratio of AL:GS showed contrasting performances in terms of compressive strength. With a higher ratio of AL:GS, finer and smoother fly ash particles could overcome obstacles caused by large and angular FCC residue particles that filled the internal voids inside the concrete structure. Therefore, pores and unreacted fly ash particles in AL:GS = 0.65 mixtures were less than AL:GS = 0.5 mixtures, leading to improved compressive strength.

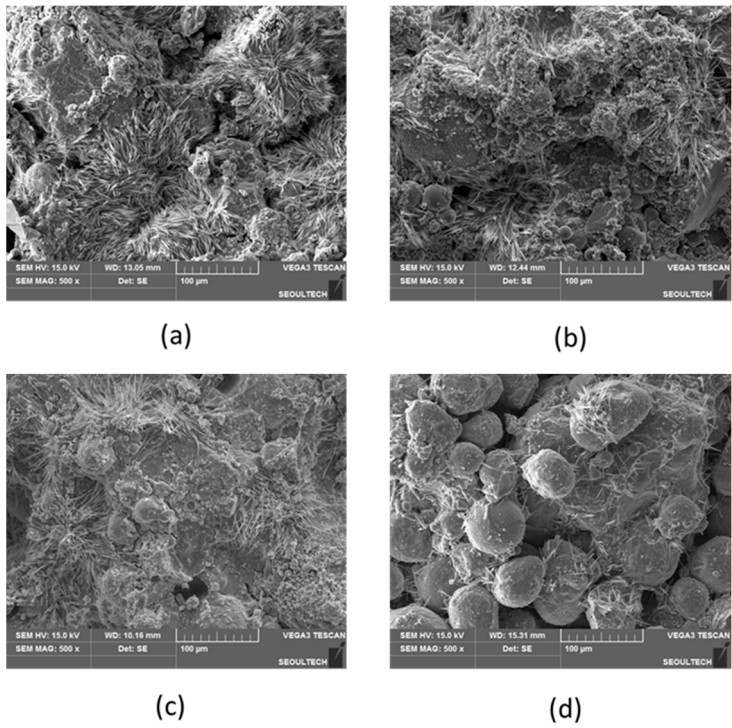

**Figure 8.** SEM images of geopolymer concrete specimens. (**a**) G1; (**b**) G5; (**c**) G7; (**d**) G11.

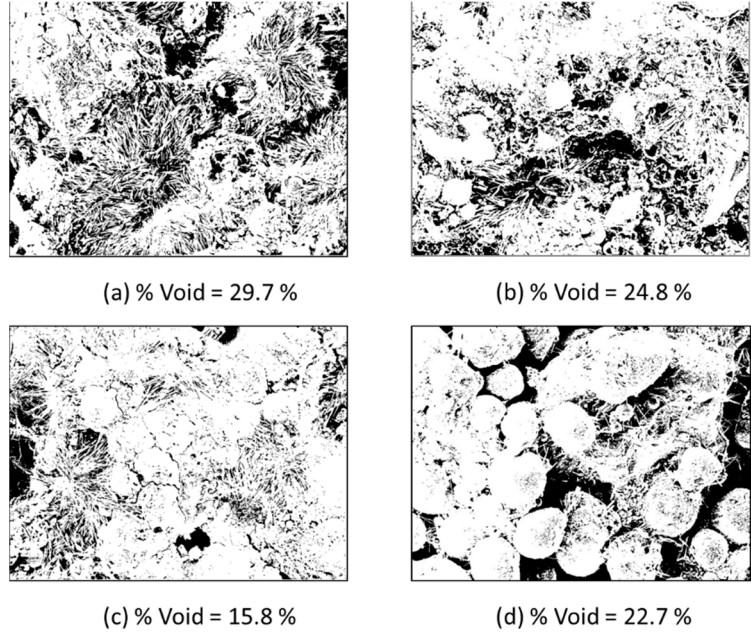

**Figure 9.** Percentage of voids in binary SEM images of geopolymer concrete specimens after image acquisition procedures. (**a**) G1; (**b**) G5; (**c**) G7; (**d**) G11.

## 4. Conclusions

In this study, experimental results provided a detailed insight into rheological and mechanical behaviors of geopolymer concrete comprising fly ash and FCC residue as aluminosilicate precursors. Key findings from this study are summarized as follows:

1.  With low rheological properties such as yield stress and plastic viscosity, fly ash-based geopolymer concrete showed better workability compared to specimens comprising FCC residue. With the presence of large and angular-shaped FCC residue particles, fresh geopolymer concrete containing both fly ash and FCC residue showed up to 12% and 39% higher yield stress and plastic viscosity compared to that consisting of only fly ash. Therefore, for the incorporation of FCC residue into geopolymer concrete, there was a need for extra alkali liquid to maintain similar workability to fly ash-based geopolymer concrete.

2.  The 7-day compressive strength, elastic modulus, and flexural strength of geopolymer concrete investigated in this study varied within 10–26 MPa, 12.3–21.3 GPa, and 0.9–2.8 MPa, with percentages of FCC residue content ranging from 0–100% by mass as replacement of fly ash. In the case of AL:GS ratios of 0.40 and 0.50, mechanical properties (i.e., compressive strength, elastic modulus, and flexural strength) of geopolymer concrete decreased with an increase in FCC residue content. Meanwhile, this trend was opposite with AL:GS ratio of 0.65 where geopolymer performance increased with an increase in FCC residue content. Apart from the mixture with AL:GS = 0.65, with partial replacement of fly ash, FCC residue with large and angular-shaped particles created more voids inside the concrete structure and resulted in mechanical properties reduction.

3.  SEM observations showed that the microstructure of geopolymer concrete consisting of FCC residue was more porous than fly ash-based geopolymer concrete, leading to reduced compressive strength.

4.  XRD patterns of geopolymer concrete revealed that with the replacement of FCC residue, final products from the geopolymerization of geopolymer concrete containing FCC residue were not much different compared to fly ash-based geopolymer concrete.

In conclusion, FCC residue containing high contents of alumina and silica can play the same role as fly ash in geopolymer products. From this fact, FCC residue can be partially or fully replaced with fly ash for producing geopolymer concrete.

**Author Contributions:** Conceptualization, T.A.L. and K.T.N.; funding acquisition, T.A.L.; investigation, K.T.N. and S.H.L.; methodology, T.N.N.; project administration, K.T.N.; supervision, T.A.L. and K.T.N.; visualization, S.H.L.; writing—original draft, K.T.N.; writing—review and editing, T.N.N. and S.H.L. All authors have read and agreed to the published version of the manuscript.

**Funding:** This research was funded by Vietnam National University Ho Chi Minh City (VNU-HCM) under grant number B2020-20-01.

**Institutional Review Board Statement:** Not applicable.

**Informed Consent Statement:** Not applicable.

**Data Availability Statement:** Not applicable.

**Conflicts of Interest:** The authors declare no conflict of interest.

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
