# Peer review of "Assessment of the Rheological and Mechanical Properties of Geopolymer Concrete Comprising Fly Ash and Fluid Catalytic Cracking Residue as Aluminosilicate Precursor"

_applsci, doi:10.3390/app11073032_

Round 1

Reviewer 1 Report

The article entitled: “Assessment of the rheological and mechanical properties of geopolymer concrete comprising fly ash and fluid catalytic cracking residue as aluminosilicate precursor” in line with the Applied Sciences journal. The article based on original research. It is well organized, however it requires some improvements and clarifications:

- Authors (line 5): please verify numbers is “7” instead of “5”;

- Featured Application (lines 14-15): please fill in the potential application in construction industry or remove;

- Abstract (line 16): incorrect font size;

- Introduction (lines: 40-42): please compare environmental aspects:

  • Silva G., Kim S., Aguilar R., Nakamatsu J. Natural fibers as reinforcement additives for geopolymers – A review of potential eco-friendly applications to the construction industry, Sustainable Mat. Techn. 23 (2020) e00132.
  • Korniejenko, K.; Lin, W.-T.; Šimonová, H. Mechanical Properties of Short Polymer Fiber-Reinforced Geopolymer Composites. J. Compos. Sci. 4 (2020) 128.
  • Bumanis, G., Vitola, L., Pundiene, I., Sinka, M., Bajare D. Gypsum, Geopolymers, and Starch—Alternative Binders for Bio-Based Building Materials: A Review and Life-Cycle Assessment. Sustainability 12 (2020) 5666.

- Introduction – line 56: please add abbreviation in brackets after “fluid catalytic cracking”;

- Introduction – please stress the novelty aspect of provided research;

- Materials (line 96) - please add information about the origin of the raw materials;

- Materials (lines 101-102, 106, 119, 132) – please correct the references (table, figure);

- Specimen preparation: what kind of specimens have been prepared? (for example: dimensions)

- Test methods: lack of information about the number of the samples in strength tests;

- Test methods: lack of information about SEM;

- Test methods: please add more detailed description of XRD;

- Results (lines 172, 183, 184, 197, 198, 206 and many others) – please correct the references (table, figure);

- Figure 9 – Could you discuss this results with literature?

- Results – the discussion with literature is very limited; please add wider discussion for the presented topic; compare received results and give some proposition of practical application for the investigated materials;

- References: please verify and add necessary information, for example ref. [1] – lack of the name of  journal.

Reviewer 2 Report

*)  At the end of the Introduction, please enter the description of the organization of the rest of the paper.

*) Sometimes in the text the sentence " Error! Reference 101
source not found." compares. Please, correct the mistake.

*) Some formulas are certainly not original so it is necessary to associate each of them with a relevant bibliographic reference.

*) Surely the thresholding technique allows to easily highlight the percentage of voids present in the sampled image. But what if the image is affected by uncertainty and / or inaccuracy? It is evident that it is necessary to exploit techniques based on soft compting centered on fuzzy logic or neuro-fuzzy approaches in order to well manipulate these uncertainties and / or inaccuracies. Notwithstanding that such a discussion requires an effort that goes beyond the present work (but which can be taken into consideration for future developments of the research in progress), I advise the authors to insert in the text at least one sentence that highlights this possibility. putting the following relevant works in the bibliography:

doi: 10.1007/s40815-020-01030-5

doi: 10.1007/s11042-020-08699-8

*) The conclusions are interesting and offer good food for thought for future developments. However, I advise the authors to propose at least two future lines of research to increase the interest of the scientific community.
